# Adiposity may confound the association between vitamin D and disease risk – a lifecourse Mendelian randomization study

**Tom G Richardson\*, Grace M Power, George Davey Smith**

MRC Integrative Epidemiology Unit (IEU), Population Health Sciences, Bristol Medical School, University of Bristol, Bristol, United Kingdom

## Abstract

**Background:** Vitamin D supplements are widely prescribed to help reduce disease risk. However, this strategy is based on findings using conventional epidemiological methods which are prone to confounding and reverse causation.

**Methods:** In this short report, we leveraged genetic variants which differentially influence body size during childhood and adulthood within a multivariable Mendelian randomization (MR) framework, allowing us to separate the genetically predicted effects of adiposity at these two timepoints in the lifecourse.

**Results:** Using data from the Avon Longitudinal Study of Parents and Children (ALSPAC), there was strong evidence that higher childhood body size has a direct effect on lower vitamin D levels in early life (mean age: 9.9 years, range = 8.9–11.5 years) after accounting for the effect of the adult body size genetic score (beta = −0.32, 95% CI = −0.54 to –0.10, p=0.004). Conversely, we found evidence that the effect of childhood body size on vitamin D levels in midlife (mean age: 56.5 years, range = 40–69 years) is putatively mediated along the causal pathway involving adulthood adiposity (beta = −0.17, 95% CI = −0.21 to –0.13, p=4.6 × $10^{-17}$).

**Conclusions:** Our findings have important implications in terms of the causal influence of vitamin D deficiency on disease risk. Furthermore, they serve as a compelling proof of concept that the timepoints across the lifecourse at which exposures and outcomes are measured can meaningfully impact overall conclusions drawn by MR studies.

**Funding:** This work was supported by the Integrative Epidemiology Unit which receives funding from the UK Medical Research Council and the University of Bristol (MC_UU_00011/1).

\*For correspondence:
Tom.G.Richardson@bristol.ac.uk

## Editor's evaluation

This paper is of great interest to readers in the fields of vitamin D and obesity. It uses a Mendelian randomization framework to separate the genetically predicted effects of adiposity at two time-points in the lifecourse, childhood, and adulthood. The key claims are well supported by the data. Higher childhood body size had a direct effect on lower vitamin D levels in early life, while in midlife, childhood body size impacted adult obesity to result in lower vitamin D levels.

## Introduction

Associations between vitamin D deficiency and disease risk have been widely reported by conventional epidemiological studies, including diseases which typically have a late-onset over the lifecourse, such as coronary artery disease, but also those which may be diagnosed in early life such as type 1 diabetes (T1D). As a result, vitamin D supplements are widely prescribed with an estimated 18%

of adults in the USA reportedly taking supplements daily (*Rooney et al., 2017*). However, there is increasing evidence from the literature suggesting that vitamin D supplements may be ineffective at reducing disease risk in the population (*Bouillon et al., 2022*). Furthermore, these conventional association studies may have been susceptible to bias, given that vitamin D levels are known to differ amongst individuals based on various lifestyle factors, including their body mass index (BMI), as well as being prone to reverse causation, for example due to inflammatory processes which are known to lower vitamin D levels (*Preiss and Sattar, 2019*).

An approach to mitigate the influence of these sources of bias is Mendelian randomization (MR), a causal inference technique which exploits the random allocation of genetic variants at birth to evaluate the genetically predicted effects of modifiable exposures on disease outcomes and circulating biomarkers (*Davey Smith and Ebrahim, 2003*, *Richmond and Davey Smith, 2022*). For example, MR has been applied in recent years to suggest that vitamin D supplements are unlikely to have a beneficial effect on risk of T1D (*Manousaki et al., 2021*), whereas a recent study suggests that genetically lowered vitamin D levels may increase the risk of multiple sclerosis (*Mokry et al., 2015*). We previously extended the application of MR to investigate epidemiological hypotheses in a lifecourse setting (known to as 'lifecourse MR'), by deriving sets of genetic variants to separate the independent effects of body size during childhood and adulthood within a multivariable framework (*Richardson et al., 2020*). Applying this approach has highlighted the putative causal role that early life adiposity may have on outcomes such as T1D risk (*Richardson et al., 2022*) and cardiac structure (*O'Nunain et al., 2022*). In contrast, we have demonstrated that its influence on other disease outcomes (e.g. cardiovascular disease; *Power et al., 2021*) is likely attributed to the long-term consequence of remaining overweight into later life.

Whilst these applications serve as powerful examples of lifecourse MR as an approach to separate the effects of the same exposure based on data derived from early and later life, it has yet to be applied to the same outcome when measured at different timepoints in the lifecourse. In this study, we applied lifecourse MR to investigate the independent effects of childhood and adult body size on 25-hydroxyvitamin D (25OHD) levels measured during childhood (mean age: 9.9 years, range = 8.9–11.5 years) using individual-level data from the Avon Longitudinal Study of Parents and Children (ALSPAC) (*Boyd et al., 2013*; *Fraser et al., 2013*) and during adulthood (mean age: 56.5 years, range = 40–69 years) using summary-level data based on individuals from the UK Biobank (UKB) study (*Manousaki et al., 2020*).

## Materials and methods
### Genetic instruments for childhood and adult body size
Derivation of genetic instruments for childhood and adulthood body size have been described in detail previously (*Richardson et al., 2020*). In brief, genome-wide association studies (GWAS) were conducted on 463,005 UKB participants (mean age: 56.5 years, range = 40–69 years) who had both reported their body size at age 10 as well as had their BMI clinically measured. Genetic instruments were identified from these analyses (based on $p < 5 \times 10^{-8}$) and the resulting genetic score for childhood body size has been validated using measured childhood BMI in ALSPAC (*Richardson et al., 2020*), the Young Finns Study (*Richardson et al., 2021*), and the Trøndelag Health (HUNT) study (*Brandkvist et al., 2020*).

### The Avon Longitudinal Study of Parents and Children
ALSPAC is a population-based cohort investigating genetic and environmental factors that affect the health and development of children. The study methods are described in detail elsewhere (*Boyd et al., 2013*; *Fraser et al., 2013*). In brief, 14,541 pregnant women residents in the former region of Avon, UK, with an expected delivery date between April 1, 1991, and December 31, 1992, were eligible to take part in ALSPAC. Detailed phenotypic information, biological samples, and genetic data which have been collected from the ALSPAC participants are available through a searchable data dictionary (http://www.bris.ac.uk/alspac/researchers/our-data/). Written informed consent was obtained for all study participants. Ethical approval for this study was obtained from the ALSPAC Ethics and Law Committee and the Local Research Ethics Committees. Measures of 25OHD levels

were obtained from non-fasting blood samples taken from ALSPAC participants at mean age 9.9 years (range = 8.9–11.5 years) which were log-transformed to ensure normality.

## Adulthood estimates of 25OHD levels

Genetic estimates on adulthood 25OHD were obtained from a previously conducted GWAS in UKB by *Manousaki et al., 2020*, as well as by *Revez et al., 2020*. All data on adulthood 25OHD analysed in this study were based on summary-level information and therefore the relevant ethical approval for the individual-level data analyses which generated them can be found in their corresponding articles. Despite 431,074 of the 476,169 participants with measures of childhood and adult body size in UKB also having measures of vitamin D levels (90.5%), there was little evidence of inflated type 1 error rates (based on the calculator at https://sb452.shinyapps.io/overlap) (*Burgess et al., 2016*). Genetic estimates were harmonized using the 'TwoSampleMR' R package (*Hemani et al., 2018*) which by default removes palindromic variants and attempts to identify proxies for any instruments whose estimates are not available in the outcome dataset.

## Lifecourse MR analysis

MR analyses to estimate genetically predicted effects of adiposity on childhood 25OHD were conducted in a one-sample setting using individual-level data from ALSPAC after generating genetic risk scores for our body size instruments with adjustment for age and sex. MR analyses to estimate effects on adulthood 25OHD were undertaken in a two-sample setting using the inverse variance weighted method (*Burgess et al., 2013*), as well as the weighted median and MR-Egger methods (*Bowden et al., 2015*; *Bowden et al., 2016*; *Supplementary file 1*—Table 1). Multivariable MR analyses were performed in one- and two-sample settings respectively for childhood and adulthood measures of 25OHD (*Sanderson et al., 2019*; *Supplementary file 1*—Table 2).

## Results

Firstly, using data measured at mean age 9.9 years in the lifecourse from the ALSPAC study, our analyses indicated that childhood body size directly influences vitamin D levels during childhood after accounting for the adult body size genetic instrument in our model (beta = −0.32 standard deviation change in natural log-transformed 25OHD per change in body size category, 95% CI = −0.54 to –0.10, p=0.004) (*Figure 1*). Using data from the UKB study, evidence of an effect of higher childhood body size on adulthood measured 25OHD levels (beta = −0.14, 95% CI = −0.10 to –0.03, p=2.4 × 10⁻⁴) attenuated in a multivariable setting upon accounting for adulthood body size (beta = 0.04, 95% CI = −0.01 to 0.08, p=0.10). In contrast, strong evidence of an effect for higher adult body size on lower 25OHD levels measured in adulthood was found in the multivariable model accounting for the childhood body size score (beta = −0.17, 95% CI = −0.21 to –0.13, p=4.6 × 10⁻¹⁷) (*Figure 1C*). This suggests that childhood body size has an indirect influence of 25OHD levels in adulthood, likely due to its persistent effect throughout the lifecourse (*Figure 1D*). The full sets of univariable and multivariable estimates derived in this analysis can be found in *Supplementary file 1*—Table 1 and *Supplementary file 1*—Table 2, respectively. Similar conclusions were drawn using estimates derived on adult 25OHD levels using findings by Revez et al (*Revez et al., 2020*; *Supplementary file 1*—Table 3).

## Discussion

Our findings suggest that increased adiposity exerts a strong effect on lower vitamin D levels during both childhood and adulthood. Separating causal from confounding factors, particularly in a lifecourse setting, would have been extremely challenging to disentangle without the use of genetic variants as achieved in this study. In contrast, appropriately accounting for confounding factors in an conventional epidemiological setting is notoriously challenging, with previous studies reporting evidence of association between vitamin D and T1D in early life even after adjusting for factors such as birthweight (*Hyppönen et al., 2001*). Taken together with evidence from previous MR studies, which have found that childhood body size (*Richardson et al., 2022*), but not vitamin D levels (*Manousaki et al., 2021*), increases risk of T1D, our results suggest that adiposity may have acted as a confounding factor on the observed association between vitamin D and T1D. Evidence from this study therefore highlights

the importance of developing a deeper understanding into the role that confounding factors, such as adiposity, may potentially have in distorting observational associations between vitamin D levels and disease risk. Future studies are therefore warranted to disentangle the causal factors which influence disease risk from non-causal confounding factors through the triangulation of multiple lines of evidence including those identified by robustly conducted MR studies.

Our conclusions are also supported by the outcomes of recent large-scale randomized controlled clinical trials (RCTs) of vitamin D supplementation. For instance, as highlighted by recent reviews (**Ebeling, 2022**; **Bouillon et al., 2022**), whilst observational studies have found that low vitamin D levels associate with increased risk of cardiovascular disease and cancer outcomes, there has been little convincing evidence from RCTs that a causal relationship may underly these findings. Additionally, recent RCTs have reported evidence of a potential interaction between vitamin D supplementation and BMI, such as the Finnish Vitamin D Trial (FIND) (**Virtanen et al., 2022**) and D-Health (**Waterhouse et al., 2021**) studies.

Effect estimates derived from MR studies are conventionally interpreted as 'lifelong effects' given that the germline genetic variants harnessed by this approach as instrumental variables are typically fixed at conception. The results of this study serve as a compelling demonstration that the timepoint at which both exposures and outcomes are measured across the lifecourse can meaningfully impact conclusions drawn by MR investigations. Whilst the childhood and adult body size genetic scores were used in this study as a proof of concept, our findings also help to further validate their utility in

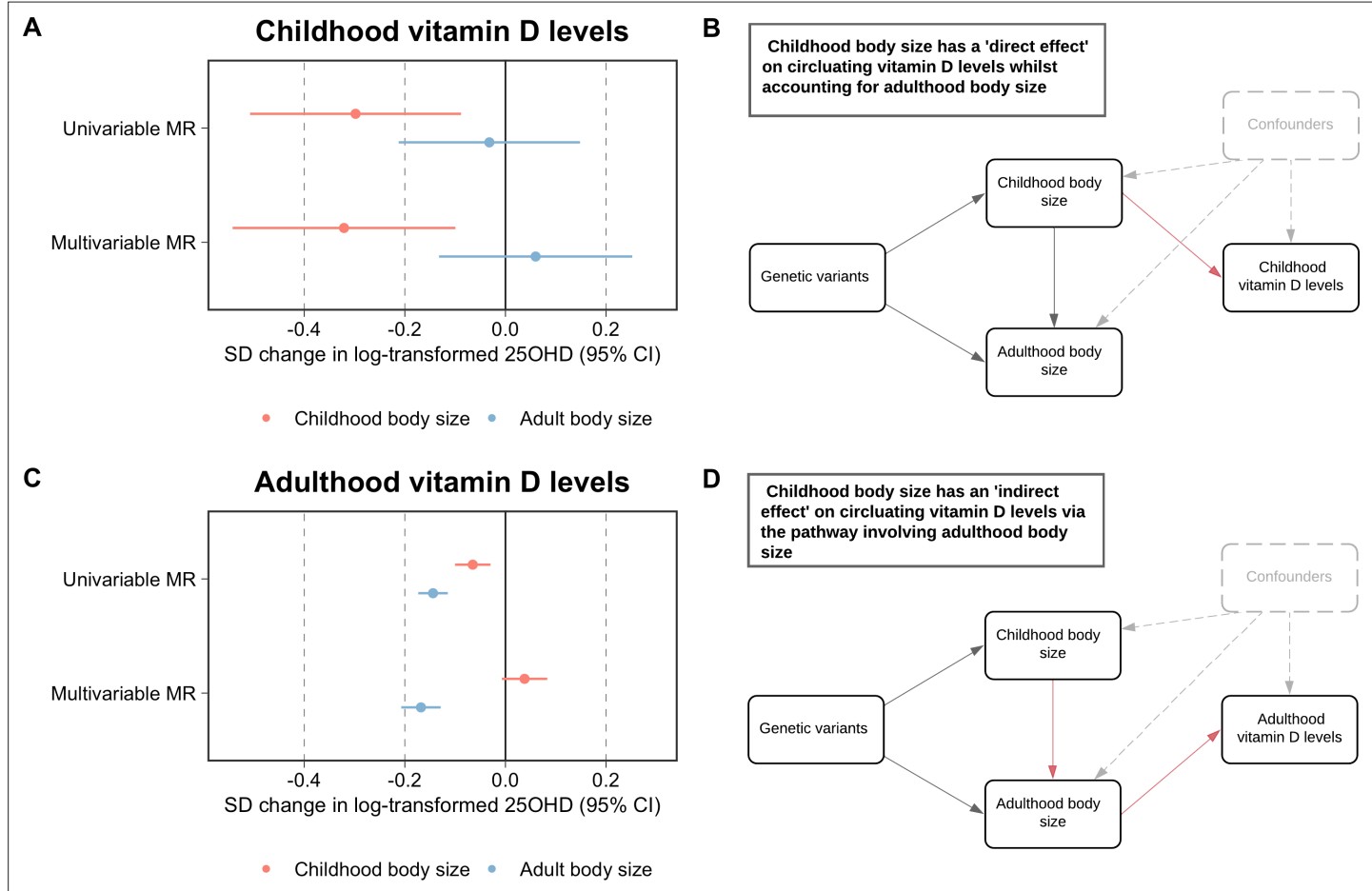

**Figure 1.** Forest plots and schematic diagrams depicting the findings from this study. (**A**) A forest plot illustrating the direct effect of childhood body size on circulating vitamin D levels measured during childhood (mean age: 9.9 years, n=3,800 participants) using data from the Avon Longitudinal Study of Parents and Children (ALSPAC) with the corresponding schematic diagram for this finding being located in panel (**B**). (**C**) A forest plot depicting the indirect effect of childhood body size on adulthood measured vitamin D levels using data from the UK Biobank study (mean age: 56.5 years, n=401,460 participants) as described in the schematic diagram presented in panel (**D**). MR – Mendelian randomization. All estimates underlying the forest plots can be found in **Supplementary file 1**.

separating the temporal effects of adiposity within a lifecourse MR framework. Overall, these findings emphasise the importance of future MR studies taking further consideration into the age of participants that their genetic estimates are based on, as well as the age at which cases are diagnosed for disease outcome estimates, before interpreting and drawing conclusions from their findings.

Furthermore, our results suggest that conducting GWAS on populations of different age groups will add value in helping uncover time-varying genetic effects scattered throughout the human genome. Findings from these endeavours should facilitate studies applying techniques such as lifecourse MR, which can provide insight into the direct and indirect effects of modifiable early life exposures on disease outcomes by harnessing genetic estimates obtained from unprecedented sample sizes when conducted in a two-sample setting. That said, lifecourse MR requires careful examination of genetic instruments to ensure that they are capable of robustly separating the effects of an exposure at different timepoints over the lifecourse (*Sanderson et al., 2022*). Doing so may help to elucidate the critical timepoints whereby conferred risk by these exposures on disease outcomes starts to become immutable, which has important implications for improving patient care in a clinical setting.

## Consent

Written informed consent was obtained for all study participants. Ethical approval for this study was obtained from the ALSPAC Ethics and Law Committee and the Local Research Ethics Committees.

## Acknowledgements

We are extremely grateful to all the families who took part in this study, the midwives for their help in recruiting them, and the whole ALSPAC team, which includes interviewers, computer and laboratory technicians, clerical workers, research scientists, volunteers, managers, receptionists, and nurses. The UK Medical Research Council and Wellcome (Grant ref: 217065/Z/19/Z) and the University of Bristol provide core support for ALSPAC. Consent for biological samples has been collected in accordance with the Human Tissue Act (2004). GWAS data was generated by Sample Logistics and Genotyping Facilities at Wellcome Sanger Institute and LabCorp (Laboratory Corporation of America) using support from 23andMe. This work was supported by the Integrative Epidemiology Unit which receives funding from the UK Medical Research Council and the University of Bristol (MC_UU_00011/1). This research was conducted at the NIHR Biomedical Research Centre at the University Hospitals Bristol NHS Foundation Trust and the University of Bristol. The views expressed in this publication are those of the author(s) and not necessarily those of the NHS, the National Institute for Health Research or the Department of Health. This publication is the work of the authors and TGR will serve as guarantor for the contents of this paper.

## Additional information

### Competing interests

Tom G Richardson: is employed part-time by Novo Nordisk on work outside of the work reported in this study. The other authors declare that no competing interests exist.

### Funding

| Funder | Grant reference number | Author |
| --- | --- | --- |
| Medical Research Council | MC_UU_00011/1 | George Davey Smith |
| UK Medical Research Council and Wellcome | 217065/Z/19/Z | George Davey Smith |

The funders had no role in study design, data collection and interpretation, or the decision to submit the work for publication.

### Author contributions

Tom G Richardson, Data curation, Software, Formal analysis, Visualization, Methodology, Writing – original draft, Writing – review and editing; Grace M Power, Formal analysis, Methodology, Writing

– review and editing; George Davey Smith, Conceptualization, Supervision, Funding acquisition, Methodology, Writing – review and editing

### Author ORCIDs
Tom G Richardson ⓘ http://orcid.org/0000-0002-7918-2040
Grace M Power ⓘ http://orcid.org/0000-0002-5702-7728
George Davey Smith ⓘ http://orcid.org/0000-0002-1407-8314

### Ethics
Human subjects: Written informed consent was obtained for all study participants. Ethical approval for this study was obtained from the ALSPAC Ethics and Law Committee and the Local Research Ethics Committees. All data on adulthood 25OHD analysed in this study were based on summary-level information and therefore the relevant ethical approval for the individual-level data analyses which generated them can be found in their corresponding articles.

### Decision letter and Author response
Decision letter https://doi.org/10.7554/eLife.79798.sa1
Author response https://doi.org/10.7554/eLife.79798.sa2

## Additional files

### Supplementary files
• Supplementary file 1. Tables with Mendelian randomization estimates supporting the conclusions of this study. Table 1 includes univariable Mendelian randomization estimates between lifecourse adiposity and vitamin D levels, Table 2 includes multivariable Mendelian randomization estimates between lifecourse adiposity and vitamin D levels, and Table 3 includes repeated analyses using vitamin D estimates from the study by Revez et al.

• MDAR checklist

### Data availability
All individual level data analysed in this study can be accessed via an approved application to ALSPAC (http://www.bristol.ac.uk/alspac/researchers/access/). Summary-level data on adulthood vitamin D levels can be accessed publicly via the OpenGWAS (https://gwas.mrcieu.ac.uk/). Figure 1 - raw estimates used to generate Figure 1A and Figure 1C can be found in Supplementary Data 1 & 2.

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
