## [Editor Report]

This paper is of great interest to readers in the fields of vitamin D and obesity. It uses a Mendelian randomization framework to separate the genetically predicted effects of adiposity at two timepoints in the lifecourse, childhood, and adulthood. The key claims are well supported by the data. Higher childhood body size had a direct effect on lower vitamin D levels in early life, while in midlife, childhood body size impacted adult obesity to result in lower vitamin D levels.

---

## [Decision Letter]

**Decision letter after peer review:**

Thank you for submitting your article "Associations between vitamin D and disease risk may be attributed to the confounding influence of adiposity during childhood and adulthood: a lifecourse Mendelian randomization study" for consideration by *eLife*. Your article has been reviewed by 2 peer reviewers, and I personally oversaw the evaluation in my joint role of Reviewing Editor and Senior Editor. The following individual involved in review of your submission has agreed to reveal their identity: Peter Ebeling (Reviewer #1).

The reviewers have discussed their reviews with one another and with me. I drafted this decision letter to help you prepare a revised submission. Please submit a revised version that addresses these concerns directly. Although we expect that you will address these comments in your response letter, we also need to see the corresponding revision clearly marked in the text of the manuscript. Some of the reviewers' comments may seem to be simple queries or challenges that do not prompt revisions to the text. Please keep in mind, however, that readers may have the same perspective as the reviewers. Therefore, it is essential that you attempt to amend or expand the text to clarify the narrative accordingly.

Essential revisions:

1) As recommended by the two reviewers, please include a discussion on recent trials or studies that shed light on the same topic of the interplay of BMI and vitamin D on health outcomes.

2) Please acknowledge the limitations and additional caveats pointed out by reviewer #2

3) In addition, please revise the manuscript as per the additional recommendations by both reviewers.

*Reviewer #1 (Recommendations for the authors):*

This reviewer has the following recommendations for the authors:

1. It might be worth including data from recent large RCTs showing interactions of BMI with outcomes related to vitamin D supplementation, cardiovascular events and falls, highlighted in a recent editorial (PMID: 35348579). This would support their data.

2. Examples are the FIND (PMID: 34982819) and D-Health (PMID: 34337905) studies with the specific outcomes of increased cardiovascular events and falls, respectively, in those supplemented participants with a BMI <25 kg/m2.

*Reviewer #2 (Recommendations for the authors):*

Introduction:

– Line 50-52: I would suggest replacing the Murai et al. 2021 reference by a paper providing a more general overview, as the Murai et al. 2021 publication describes a RCT for vitamin D supplementation in covid-19 patients specifically. PMID 34815552 (Bouillon et al. Nat Rev Endocrinol 2022) might be more suited here

– Line 53: Mokry et al. 2015 is a Mendelian randomisation study, which is further described in the second paragraph of the introduction (starting from line 61). It might be more suited to incorporate this reference in line 64-65.

Results:

– Line 82: 'has a largely indirect influence' needs to be removed here?

– Line 87: Manousaki et al. 2020 GWAS was used to select instrumental variables for adulthood vitamin D. I was wondering whether similar results are obtained when selecting variants from the Revez et al. 2020 GWAS (PMID 32242144) ? Both are conducted in UK Biobank, though there are small differences in sample sizes and number of variants (especially due to differences in minor allele frequency filtering).

– Line 93: Multivariate while in line 96: multivariable. Is there a difference here?

– Line 96-97: perhaps add 'upon accounting for childhood body size'? (as in line 94), for further clarification on analyses

Figure 1: Number is lacking in the legend (mean age: X years)

Materials and methods:

– Line 168-171: What was the percentage of sample overlap? Although sample overlap does not seem to influence the findings, it may be worth mentioning the numbers of sample overlap, aiding readers in interpreting the degree of overlap.

– Line 173-182: It is stated in line 159-160 that genetic instruments for childhood and adulthood body size are derived from Richardson et al. 2020. Would it be possible to add some more details on selection of instrumental variables (in line with STROBE-MR guidelines), as initially 295 variants for childhood and 557 variants for adulthood body size have been identified in Richardson et al. 2020 while respectively 253 and 488 are included in the MR analyses. Are palindromic SNPs retained? Are proxy SNPs identified for SNPs missing in the outcome data?

– Line 177-178: The two-sample MR design allowed the performance of additional analyses (weighted median and MR-Egger, beyond the inverse variance weighted method). Now it seems that the IVW method allowed the additional analyses rather than the two-sample MR design. Would it be possible to rephrase this sentence accordingly?

---

## [Author Response]

Essential revisions:1) As recommended by the two reviewers, please include a discussion on recent trials or studies that shed light on the same topic of the interplay of BMI and vitamin D on health outcomes.

We have added some discussions to address this to page 6 of the manuscript as described in detail below.

2) Please acknowledge the limitations and additional caveats pointed out by reviewer #2

We have added some discussion of page 7 to address this point directly in response to recommendation to reviewer #2 below.

Reviewer #1 (Recommendations for the authors):This reviewer has the following recommendations for the authors:1. It might be worth including data from recent large RCTs showing interactions of BMI with outcomes related to vitamin D supplementation, cardiovascular events and falls, highlighted in a recent editorial (PMID: 35348579). This would support their data.

Thank you for sharing this review with us which corroborates the findings of our study. We have discussed this as recommended on page 6 of the Discussion:

‘Our conclusions are also supported by the outcomes of recent large-scale randomized controlled clinical trials (RCTs) of vitamin D supplementation. For instance, as highlighted by a recent review (Ebeling, 2022), whilst observational studies have found that low vitamin D levels associate with increased risk of cardiovascular disease and cancer outcomes, there has been little convincing evidence from RCTs that a causal relationship may underly these findings.’

2. Examples are the FIND (PMID: 34982819) and D-Health (PMID: 34337905) studies with the specific outcomes of increased cardiovascular events and falls, respectively, in those supplemented participants with a BMI <25 kg/m2.

Evidence of an interaction between vitamin D supplementation and BMI reported by these two RCTs has now been discussed on page 6:

‘Moreover, recent RCTs have reported evidence of a potential interaction between vitamin D supplementation and BMI, such as the Finnish Vitamin D Trial (FIND) (Virtanen et al., 2022) and D-Health (Waterhouse et al., 2021) studies.’

Reviewer #2 (Recommendations for the authors):Introduction:– Line 50-52: I would suggest replacing the Murai et al. 2021 reference by a paper providing a more general overview, as the Murai et al. 2021 publication describes a RCT for vitamin D supplementation in covid-19 patients specifically. PMID 34815552 (Bouillon et al. Nat Rev Endocrinol 2022) might be more suited here

Many thanks for this suggestion. We have updated this reference as suggested.

– Line 53: Mokry et al. 2015 is a Mendelian randomisation study, which is further described in the second paragraph of the introduction (starting from line 61). It might be more suited to incorporate this reference in line 64-65.

As recommended, we have moved this reference to the MR section of this work (Page 3):

‘For example, MR has been applied in recent years to suggest that vitamin D supplements are unlikely to have a beneficial effect on risk of type 1 diabetes (Manousaki et al., 2021), whereas a recent study suggests that genetically lowered vitamin D levels may increase risk of multiple sclerosis (Mokry et al., 2015).’

Results:– Line 82: 'has a largely indirect influence' needs to be removed here?

Thank you. We have now removed this text.

– Line 87: Manousaki et al. 2020 GWAS was used to select instrumental variables for adulthood vitamin D. I was wondering whether similar results are obtained when selecting variants from the Revez et al. 2020 GWAS (PMID 32242144) ? Both are conducted in UK Biobank, though there are small differences in sample sizes and number of variants (especially due to differences in minor allele frequency filtering).

We have repeated our analysis using the Revez et al. GWAS results (unadjusted for BMI) and added results to Supplementary Table 3. These results did not change the overall conclusions of this paper as reported on page 4:

‘Similar conclusions were drawn using estimates derived on adult 25OHD levels using findings by Revez et al. (Revez et al., 2020) (Supplementary Table 3).’

– Line 93: Multivariate while in line 96: multivariable. Is there a difference here?

To be consistent we have now changed ‘multivariate’ to ‘multivariable’ on line 93.

– Line 96-97: perhaps add 'upon accounting for childhood body size'? (as in line 94), for further clarification on analyses

This sentence has now been updated to incorporate your suggestion:

‘In contrast, strong evidence of an effect for higher adult body size on lower 25OHD levels measured in adulthood was found in the multivariable model accounting for the childhood body size score (Β=-0.17, 95% CI=-0.21 to -0.13, P=4.6x10^-17^) (Figure 1C).’

Figure 1: Number is lacking in the legend (mean age: X years)

Thank you for spotting this – we have now updated the text to say ‘mean age 56.5 years’

Materials and methods:– Line 168-171: What was the percentage of sample overlap? Although sample overlap does not seem to influence the findings, it may be worth mentioning the numbers of sample overlap, aiding readers in interpreting the degree of overlap.

Although it is challenging to determine the exact percentage of sample overlap in the GWAS of our exposure variables compared with the vitamin D GWAS undertaken by Manousaki et al., we have estimated the proportion of overlap using individual level from UKB on page 7:

‘Despite 431,074 of the 476,169 participants with measures of childhood and adult body size in UKB also having measures of vitamin D levels (90.5%), there was little evidence of inflated type 1 error rates (based on the calculator at https://sb452.shinyapps.io/overlap) (Burgess et al., 2016)

– Line 173-182: It is stated in line 159-160 that genetic instruments for childhood and adulthood body size are derived from Richardson et al. 2020. Would it be possible to add some more details on selection of instrumental variables (in line with STROBE-MR guidelines), as initially 295 variants for childhood and 557 variants for adulthood body size have been identified in Richardson et al. 2020 while respectively 253 and 488 are included in the MR analyses. Are palindromic SNPs retained? Are proxy SNPs identified for SNPs missing in the outcome data?

We have added the following to the Methods section on page 8 to add further clarity to our instrument selection:

‘Genetic estimates were harmonised using the ‘TwoSampleMR’ R package (Hemani et al., 2018) which by default removes palindromic variants and attempts to identify proxies for any instruments whose estimates are not available in the outcome dataset.’

– Line 177-178: The two-sample MR design allowed the performance of additional analyses (weighted median and MR-Egger, beyond the inverse variance weighted method). Now it seems that the IVW method allowed the additional analyses rather than the two-sample MR design. Would it be possible to rephrase this sentence accordingly?

Thank you – this sentence on page 8 has now been rephrased accordingly:

‘MR analyses to estimate effects on adulthood 25OHD were undertaken in a two-sample setting using the inverse variance weighted (IVW) method (Burgess et al., 2013), as well as the weighted median and MR-Egger methods (Bowden et al., 2015, Bowden et al., 2016) (Supplementary Table 1).’